# Enhancing Protein Language Model with Feature Integration for Anticancer Peptide Prediction

Tiara Natasha Binte Sayuti[1], Shen Cheng[2], Santhisenan Ajith[1], Abdul Hadi Bin Abdul Samad[3], and Jagath C. Rajapakse[1]

*Abstract*—In the fight against cancer, anticancer peptides (ACP) hold promising therapeutic potential due to their selective cytotoxicity and lower side effects compared to traditional treatments. However, identifying novel ACP is challenged by high costs and labor-intensive processes. Protein language models (PLMs), such as ESM-2 and ProtBERT, have revolutionized peptide prediction by leveraging vast datasets to capture complex biological patterns through pre-training. However, they often struggle to accurately model specific biochemical interactions. To address this limitation, we integrated four sequence-based features: amino acid composition (AAC), dipeptide composition (DPC), composition of k-spaced amino acid group pairs (CKS), and k-mer sparse matrix (k-mer) through a cross-attention mechanism. These features infuse biochemical insights that PLM alone may overlook, enabling a more detailed prediction of anticancer properties. This integration enhances biochemical insights, improving prediction accuracy by 15.8% for ProtBERT and 2.9% for ESM-2, with ESM-2 achieving the highest accuracy at 77.8%. SHapley Additive exPlanations (SHAP) analysis confirms the importance of these features, demonstrating that incorporating amino acid features into PLMs enhances ACP prediction.

*Index Terms*—amino-acid compositions, anticancer peptide prediction, cross-attention, protein language models

## I. INTRODUCTION

Cancer remains a daunting global health concern. In 2022, the World Health Organization (WHO) reported 20 million new cancer cases and 9.7 million fatalities, translating to approximately 1 in 5 individuals developing cancer within their lifetime. By 2050, over 35 million new diagnoses are forecasted, emphasizing the urgent need for enhanced cancer prevention, detection, and treatment efforts [1].

Traditional treatments like chemotherapy and radiotherapy are central to current cancer management. However, their effectiveness wanes as tumors develop resistance, and the side effects significantly impact patients' quality of life [2]. When these treatments end, alternatives are often scarce, presenting a substantial clinical challenge. Thus, exploring alternatives with comparable efficacy and few side effects are crucial.

*Research supported by AcRF Tier-1 grant RG14/23 of Ministry of Education, Singapore.

[1]Tiara Natasha Binte Sayuti, Santhisenan Ajith, and Jagath C. Rajapakse are with the Health Informatics Lab from the College of Computing and Data Science, Nanyang Technological University, Singapore. (E-mails: s230112@e.ntu.edu.sg, santhise001@e.ntu.edu.sg, asjagath@ntu.edu.sg)

[2]Shen Cheng is with the College of Computing and Data Science, Nanyang Technological University, Singapore. (E-mail: s230143@e.ntu.edu.sg)

[3]Abdul Hadi Bin Abdul Samad is with the School of Electrical and Electronic Engineering, Nanyang Technological University, Singapore. (E-mail: e230080@e.ntu.edu.sg)

Anticancer peptides (ACP) are promising alternatives due to their selective cytotoxicity towards cancer cells and relatively lower side effects. ACP offer a safer, more targeted approach, making them appealing as potential replacements for traditional therapies [3]. However, discovering and validating new ACP face challenges such as high synthesis costs and extensive laboratory efforts required for experimental validation, which are resource-intensive and time-consuming [4].

Recent advancements in protein language models (PLM) offer significant advantages for ACP prediction. PLMs leverage vast amounts of protein sequence data to understand and predict protein structure, function, and interactions, providing rich representations of peptide sequences [5]–[7]. PLM are suitable for this task due to their ability to capture intricate relationships within protein sequences through pre-training on large datasets, followed by fine-tuning for specific tasks [5]. Despite their sophisticated ability to encode protein sequences into informative embeddings, PLM exhibit limitations in capturing specific biochemical interactions, making it necessary to enhance the model through additional features.

To further enhance the predictive power of ACP predictors, we incorporate four sequence-based features: amino acid compositions (AAC), dipeptide compositions (DPC), compositions of k-spaced amino acid group pairs (CKS), and k-mer sparse matrix (k-mer) with the PLM embeddings. These features provide biochemical insights that complement the embeddings from PLMs, enabling a more detailed and accurate prediction of anticancer properties. AAC provides an overview of the peptide's composition, DPC captures local sequence order, CKS identifies non-adjacent relationships, and k-mer represents specific sub-sequences of amino acids. Together, these features offer improved representation of ACPs and differentiate the method from previous approaches, potentially accelerating the discovery of new therapeutic peptides.

## II. RELATED WORKS

### A. Machine Learning in ACP Prediction

Early studies in ACP prediction predominantly utilized traditional machine learning models. For instance, Vijayakumar and Lakshmi developed a prediction technique using Support Vector Machines based on compositional and distributional metrics of amino acids, significantly improving the precision of ACP prediction [8]. On the other hand, Chen et al. improves ACP predictor with optimized g-gap dipeptides [9].

## B. Deep Learning in ACP Prediction

With the advent of deep learning, neural network-based models began to show improved performance in ACP prediction. Yuan et al. proposed a model that ensembles deep learning and machine learning algorithms, specifically integrating bidirectional long short-term memory (BiLSTM), Convolutional Neural Network (CNN), and Light Gradient Boosting Machine (LightGBM) to predict ACPs from positional encoded peptide sequences [10]. Yang et al. also developed a deep learning framework called CACPP, that utilizes CNN and contrastive learning to enhance the accuracy of ACP predictions [11].

## C. Protein Language Model in ACP Prediction

Recent studies have shifted focus towards integrating general word embeddings to represent peptide sequences. Zhu et al. introduced ACP-ST, an ACP prediction model based on general word embedding features and a Swin Transformer with a multi-head self-attention mechanism, achieving superior performance on benchmark datasets [12]. More recent advancements have leveraged PLMs and further revolutionized the field. Bepler and Berger discussed how deep protein language models can learn evolutionary, structural, and functional information from vast protein sequence databases, capturing complex biological insights that can inform function predictions and design new proteins [13]. Additionally, Ruffolo and Madani discussed the foundations and applications of PLMs in protein engineering. They emphasized how PLMs, trained on extensive datasets of protein sequences, can learn the underlying patterns of protein structure and function, enabling tasks such as sequence design and variant effect prediction [14].

## D. Incorporating Sequence-Based Features in ACP Prediction

Incorporating sequence-based features has proven to be a positive component in enhancing predictive models. Fazal et al. presented an approach using kernel sparse representation classification with the composition of k-spaced amino acid pairs to capture a diverse range of peptide sequences, yielding a comprehensive feature vector that enhances ACP classification performance [15]. Similarly, Li et al. employed sequence-based features such as amino acid composition, dipeptide composition, and k-spaced amino acid pairs in improving the predictive power of ACP prediction, demonstrating the importance of sequence-based features in enhancing prediction models [16].

## E. Our Motivation

PLMs effectively learn embeddings directly from sequence data, capturing rich contextual information crucial for understanding protein structure and function. However, incorporating additional features such as AAC, DPC, CKS, and k-mer provides further biochemical and structural insights. These features offer complementary information that is not captured by sequence embeddings alone, potentially enhancing the performance of PLMs. Motivated by these advancements,

our work integrates the four sequence-based features with PLM embeddings via cross-attention, aiming to generate a more enriched representation of ACPs and ultimately enhance the performance of the ACP prediction model.

## III. METHODOLOGY

### A. Dataset

We utilize the AntiCP 2.0 dataset, a collection of experimentally validated peptides for anticancer activity, including 861 anticancer and 861 non-anticancer peptides [17]. The balanced nature of the AntiCP 2.0 dataset addresses class imbalance, mitigating bias towards the more prevalent class and enhancing the performance of predictive models [17].

### B. Protein Language Models

PLMs like ProtBERT and ESM-2 apply natural language processing (NLP) techniques to analyze protein sequences, aiming to predict protein structure, function, and interactions. These models encode protein sequences into high-dimensional embeddings, akin to how NLP models grasp word context and meaning in sentences [18].

*1) ProtBERT:* ProtBERT employs the Bidirectional Encoder Representations from Transformers (BERT) architecture, utilizing masked language modeling (MLM) during pretraining. This technique involves masking random segments of protein sequences for the model to predict, facilitating the learning of contextual relationships within the sequences [6]. Trained on the Big Fantastic Dataset (BFD), ProtBERT captures diverse protein relationships directly from sequence data, enabling it to understand both local and global structural information within protein sequences, making it a versatile tool for various bioinformatics tasks [6].

*2) Evolutionary Scale Modeling 2 (ESM-2):* ESM-2 also employs MLM but distinguishes itself by leveraging evolutionary data. It is trained on the UniRef protein sequence dataset, which provides a broader representation of protein sequences through clustering similar sequences to reduce redundancy [7]. This dataset enables ESM-2 to incorporate evolutionary patterns, enhancing its ability to predict and understand protein structures and functions.

By training on evolutionary data, ESM-2 can more effectively model protein dynamics and structural nuances, which are critical for tasks such as high-resolution structure prediction. This evolutionary insight enables ESM-2 to offer a more comprehensive understanding of protein sequences compared to models that do not utilize such data [7].

While both ProtBERT and ESM-2 utilize MLM, their training datasets and focus differ. ProtBERT's training on BFD enables it to capture a wide range of protein relationships without evolutionary bias, making it highly versatile for various peptide analysis [6]. In contrast, ESM-2's training on the UniRef dataset leverages evolutionary information, providing a deeper understanding of protein dynamics and structure [7].

By integrating these complementary sequence-based features with the PLM embeddings, we enhance the model's representation of anticancer peptides. This approach combines

the deep contextual knowledge derived from PLMs with the specific biochemical insights provided by sequence-based features, ultimately improving the accuracy of ACP prediction.

*C. Sequence-Based Features*

Sequence-based features are employed to extract attributes that encode biological knowledge about peptide sequences into a format that computational models can process. These features are crafted based on structural and biochemical properties of peptides. In this study, four sequence-based features were utilised to increase the input information of the amino acids in the peptide sequences. The four features are AAC, DPC, CKS and k-mer.

*1) Amino Acid Composition:* AAC represents the relative frequency of each standard amino acid within a peptide and is given for a peptide sequence of length $L$ as:

$$AAC(a) = \frac{Count(a)}{L} \tag{1}$$

where $Count(a)$ is the number of times amino acid $a$ appears in the peptide sequence. This feature provides a 21-dimensional vector where each dimension corresponds to the relative frequency of one of the 20 standard amino acids and one dimension for the relative frequency of any non-standard or unknown amino acids. Quantifying the relative abundance of amino acids offers insight into the primary structure and key functional properties, such as hydrophobicity, charge, and isoelectric point, all of which are influenced by the side chains. These properties play a critical role in determining anticancer activity.

*2) Dipeptide Composition:* DPC quantifies the frequency of consecutive amino acid pairs throughout the peptide sequence and is given by:

$$DPC(a, b) = \frac{Count(a, b)}{L - 1} \tag{2}$$

where $Count(a, b)$ is the number of times the dipeptide $ab$ appears in a peptide sequence of length $L$. By capturing the local sequence order and interactions between adjacent amino acids, including secondary structures and potential electrostatic interactions between charged residues, this approach effectively distinguishes ACPs from non-ACPs.

*3) Composition of K-Spaced Amino Acid Group Pairs:* CKS captures long-range interactions between non-adjacent amino acids, which are essential for understanding tertiary structure and function. CKS groups amino acids based on physicochemical properties, such as aliphatic, aromatic, positive charge, negative charge, or uncharged, revealing their impact on peptide folding and stability. The idea is to count occurrences of pairs of these groups separated by $k$ residues within the sequence, providing insight into how these properties are distributed along the peptide chain.

Let $G_a$ and $G_b$ be two groups, and $k$ be the gap (i.e., the number of residues between the pair). The CKS feature for a given gap $k$ is computed as:

$$CKS(G_a, G_b, k) = \frac{\sum_{i=1}^{L-k-1} I(x_i \in G_a, x_{i+k+1} \in G_b)}{\sum_{i=1}^{L-k-1} I(x_i \in G, x_{i+k+1} \in G)} \tag{3}$$

where I (condition) is an indicator function that equals 1 if the condition is true and 0 otherwise, $x_i$ is the amino acid at position $i$, $G$ is the set of all amino acid groups, and $L$ is the length of the sequence. Values of $k$ range from zero to five are used to capture various levels of long-range interactions within the peptide, enhancing the model's understanding of how peptides fold and how their structural properties influence their ability to interact and disrupt cancer cell membranes.

*4) K-mer Sparse Matrix:* The $k$-mer capture the presence and frequency of specific $k$-length subsequences within a peptide sequence. In this work, $k = 3$ (triad) is chosen as it strikes a balance between capturing local recurring sequence motifs and maintaining computational efficiency. This choice is based on the conjoint triad feature, a bioinformatics technique that groups amino acids into seven classes (AGV, ILFP, YMTS, HNQW, RK, DE, C) based on their dipole moments and side chain volumes. This approach enhances the model's ability to recognize structural motifs that are critical for peptide biological functions, such as receptor binding, which is particularly important for ACPs.

All possible combinations of triads are generated from the group labels. Each amino acid in the peptide sequence is translated into its corresponding group label using a translation dictionary. This reduces the complexity of the sequence while preserving essential biochemical properties.

For each translated sequence, a frequency matrix is constructed where each row corresponds to a triad and each column corresponds to a position in the sequence. The value at each position in the matrix indicates the presence or absence of the corresponding triad at that position as follows:

$$M = \{m_{ij}\}_{7^k \times (L-k+1)} \tag{4}$$

$$m_{ij} = \begin{cases} 1, & \text{if } a_j a_{j+1} a_{j+2} = k\text{-mer}(i) \\ 0, & \text{else} \end{cases} \tag{5}$$

Afterwards, the frequency matrix is decomposed using Singular Value Decomposition (SVD) into three matrices: $U$, $S$, and $V^T$ defined as follows:

$$M = U \cdot S \cdot V^T \tag{6}$$

where the matrix $U$ contains the left singular vectors, $S$ is a diagonal matrix with singular values, and $V^T$ contains the right singular vectors.

The k-mer sparse matrix is constructed by combining the components captured by the left singular vectors $U$ and scaling them by their corresponding singular values $S$. This is done to capture the most significant patterns in the k-mer composition of the sequence. The k-mer sparse matrix for a sequence is given by:

$$k\text{-}mer = \sum_{i=1}^{L-K+1} \left( \frac{u_i \cdot s_i}{L} \right) \tag{7}$$

where $u_i$ are the rows of left singular vector $U$ and $s_i$ are the singular values from diagonal matrix $S$, normalized by the sequence length $L$.

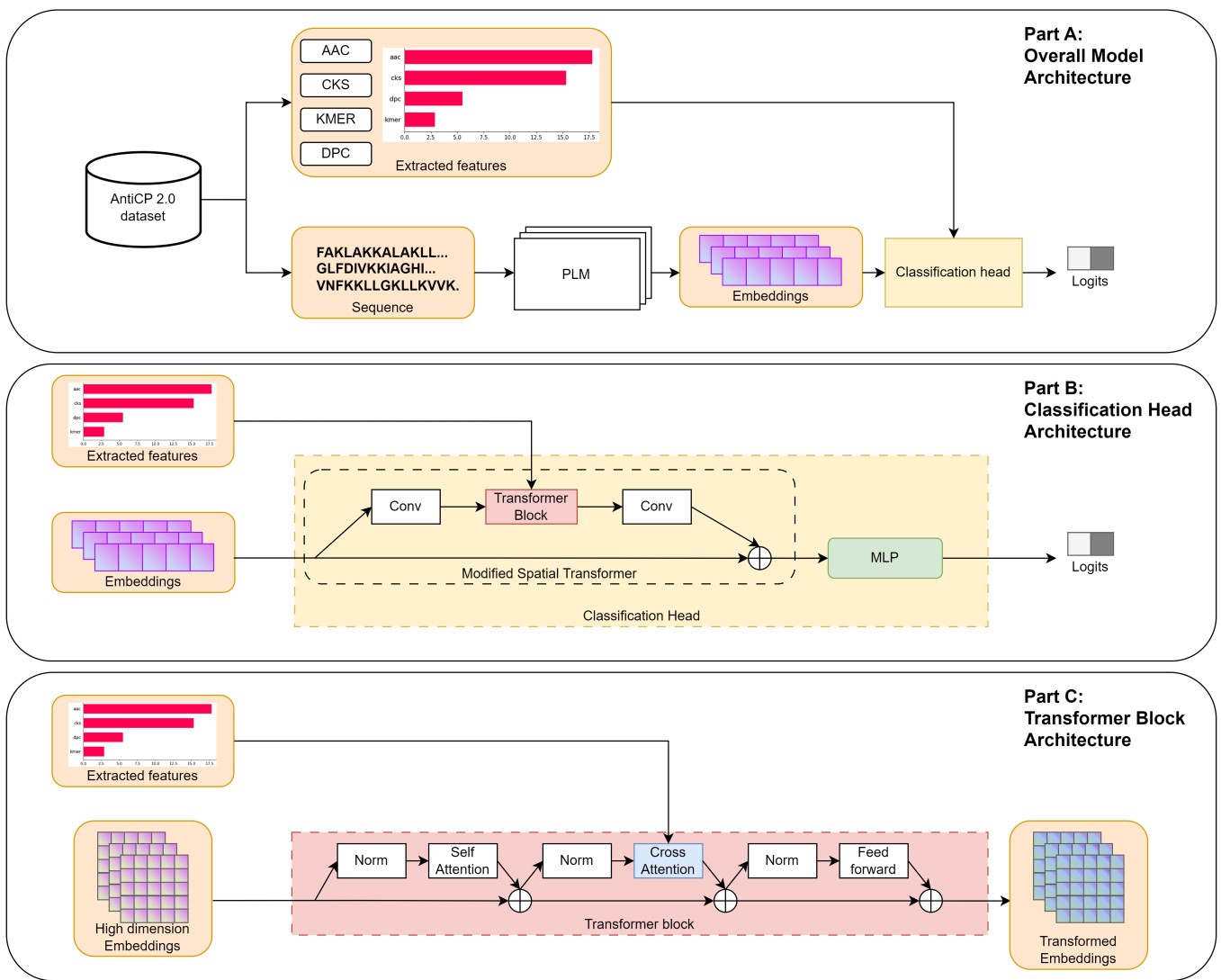

Fig. 1. Illustration of the proposed architecture for ACP prediction, including the architecture of the classifier and its constituent elements. Part A shows the overall model, from feature extraction to the classification head. Part B zooms in on the classification head, highlighting the modified spatial transformer block outlined in dashed lines, alongside an MLP block. Part C details the transformer block, emphasizing the cross-attention mechanism.

## D. Model Architecture

The proposed model architecture for ACP classification integrates sequence-based features and PLM embeddings through a classification head via cross-attention. This head includes a modified spatial transformer block [19] and a multi-layer perceptron (MLP), collectively designed to enhance the model's capability to capture and retain salient features, thereby improving overall classification performance. The model architecture is illustrated in Figure 1. Initially, the peptide sequences are processed through a PLM to generate embeddings that encapsulate the complex dependencies and contextual information inherent in the sequences, as shown in Figure 1 Part A. The classification head leverages both the sequence-based features and the PLM embeddings to produce accurate predictions. It comprises a modified spatial transformer block

and a MLP.

The modified spatial transformer block in Figure 1 Part B includes three key components: an input convolution layer, a transformer block, and an output convolution layer [19]. The input convolution layer projects the embeddings into a higher-dimensional space, facilitating enhanced feature extraction. This step is crucial for preparing the embeddings for more detailed processing in subsequent layers.

The core of the transformer block in Figure 1 Part C is decomposed into three distinct stages: self-attention, cross-attention, and feed-forward. The self-attention stage allows the model to focus on different parts of the embeddings themselves, capturing the internal structure and relationships within the sequence. The cross-attention stage integrates the sequence-based features with the embeddings, enabling the

TABLE I
PERFORMANCE COMPARISON OF FEATURE INTEGRATION TO PROTEIN LANGUAGE MODELS PROTBERT AND ESM-2

| Method | Accuracy | F1 | Precision | Sensitivity | Specificity |
|---|---|---|---|---|---|
| CNN | 0.7164 | 0.7087 | 0.7284 | 0.6901 | 0.7427 |
| Bi-LSTM | 0.6901 | 0.6728 | 0.7124 | 0.6374 | 0.7427 |
| LightGBM | 0.7398 | 0.7493 | 0.7228 | 0.7778 | 0.7017 |
| ProtBERT | 0.5380 | 0.6108 | 0.5277 | 0.7251 | 0.3509 |
| ESM-2 | 0.7485 | 0.7598 | 0.7273 | 0.7953 | 0.7018 |
| ProtBERT with feature integration | 0.6959 | 0.7263 | 0.6603 | **0.8070** | 0.5848 |
| ESM-2 with feature integration | **0.7778** | **0.7829** | **0.7654** | 0.8012 | **0.7544** |

model to leverage external contextual information effectively. This mechanism ensures that the model considers both the intrinsic properties of the sequences and the additional biochemical features. Finally, the feed-forward stage further processes the information, enhancing the model's ability to capture complex patterns.

The final convolution layer in Figure 1 Part B projects the transformed embeddings back to the original specified dimension. This processed output is then combined with the original embeddings through a residual connection, ensuring that essential features are preserved and not diminished during the transformation process. The MLP, consisting of two linear layers, serves as the final stage of the classification head. It processes the output from the modified spatial transformer block to produce the final logits used for classification.

## IV. EXPERIMENTS AND RESULTS

### A. Data Preprocessing

Effective data preprocessing is important for preparing the peptide sequences and their associated features for deep learning models. In this study, each peptide sequence was tokenized and padded to a maximum length of 50 to ensure uniformity. The maximum length of 50 was chosen as it was the length of the longest peptide in the dataset. In addition, sequence-based features such as AAC, DPC, CKS and k-mer were extracted and accurately prepared before model training.

### B. Model Training

Our peptide classification model was designed with specific hyperparameters to ensure robust performance and generalizability. We used a learning rate of $3 \times 10^{-5}$, balancing the speed of convergence and training stability. The weight decay was set to $1 \times 10^{-4}$ within the AdamW optimizer, which incorporates weight decay directly into the weight update rule. This regularization technique penalizes large weights, encouraging simpler models by reducing the risk of overfitting.

The model was trained for 20 epochs with a batch size of 8, with an early stopping mechanism in place to halt training if no improvement in validation loss was observed over 10 consecutive epochs with a minimum delta of 10. This allows for efficient training while managing memory constraints effectively. This ensured that the model generalized well to unseen data. Dropout is used to randomly drop units during training to prevent overfitting. However, in our case, the model achieved optimal results without the need for dropout,

likely due to the effective regularization already provided by weight decay and early stopping.

To further ensure the robustness and generalizability, we employed 5-fold cross-validation, where the dataset was split into five subsets, training on four and validating on the remaining one. The best model across five iterations was evaluated on the test set. This provides a comprehensive performance assessment, ensuring generalizability across different datasets and conditions.

### C. Performance Results

The performance of our model was evaluated using accuracy, F1 measure, precision, sensitivity, and specificity. Table I presents a comprehensive comparative analysis of our model with feature integration using cross-attention against a range of baseline models, including CNN, Bi-LSTM, and LightGBM. Additionally, experiments using only ESM-2 and ProtBERT embeddings for the prediction of anticancer peptides were also conducted.

The ESM-2 with feature integration using cross-attention model achieved the highest overall accuracy, registering at 0.7778, thereby surpassing all other models. This superior performance highlights the robustness and reliability of integrating additional features with cross-attention into the PLM, attesting to its efficacy in accurately classifying a majority of samples.

In terms of the F1 score, which assesses the balance between precision and sensitivity, the ESM-2 with feature integration using a cross-attention model continues to lead with a score of 0.7829. This result indicates that our model not only effectively detects positive samples but also maintains a commendably low rate of false positives.

Moreover, the same model demonstrated the highest precision among the models evaluated, with a score of 0.7654. This metric illustrates the model's capacity to identify a greater proportion of true positives, reducing the risk of false positive errors.

For sensitivity, the ESM-2 with feature integration model performed exceptionally well, achieving a score of 0.8012. This metric indicates the model's ability to correctly identify positive samples. It is noteworthy that the ProtBERT with feature integration method achieved the highest sensitivity at 0.8070, emphasizing how the integration of additional features significantly enhances the model's ability to detect true positive samples compared to using the embeddings alone.

TABLE II

ABLATION ANALYSIS OF FUSION OF DIFFERENT FEATURES CONFIGURATIONS USING A CROSS-ATTENTION MODEL

| Features | Accuracy | F1 | Precision | Sensitivity | Specificity |
|---|---|---|---|---|---|
| AAC | 0.7661 | 0.7661 | 0.7661 | 0.7661 | 0.7661 |
| DPC | 0.7222 | 0.7425 | 0.6919 | 0.8011 | 0.6433 |
| k-mer | 0.7485 | 0.7514 | 0.7429 | 0.7602 | 0.7368 |
| CKS | 0.7398 | 0.7479 | 0.7253 | 0.7719 | 0.7076 |
| AAC + DPC | 0.7456 | 0.7616 | 0.7165 | **0.8129** | 0.6783 |
| AAC + CKS | 0.7281 | 0.7320 | 0.7216 | 0.7427 | 0.7135 |
| AAC + k-mer | 0.7047 | 0.7038 | 0.7059 | 0.7018 | 0.7076 |
| DPC + CKS | 0.7281 | 0.7380 | 0.7120 | 0.7661 | 0.6901 |
| DPC + k-mer | 0.7251 | 0.7251 | 0.7251 | 0.7251 | 0.7251 |
| CKS + k-mer | 0.7749 | 0.7794 | 0.7640 | 0.7953 | 0.7544 |
| AAC + DPC + CKS | 0.7398 | 0.7375 | 0.7440 | 0.7310 | 0.7485 |
| AAC + DPC + k-mer | 0.7398 | 0.7450 | 0.7303 | 0.7602 | 0.7193 |
| AAC + CKS + k-mer | 0.7632 | 0.7652 | 0.7586 | 0.7719 | 0.7544 |
| DPC + CKS + k-mer | 0.7602 | 0.7500 | **0.7834** | 0.7193 | **0.8012** |
| AAC + DPC + CKS + k-mer | **0.7778** | **0.7829** | 0.7654 | 0.8012 | 0.7544 |

The highest specificity was observed in the ESM-2 with feature integration model at 0.7544, showcasing its ability to correctly identify negative samples.

Feature integration with cross-attention mechanisms has not only bolstered the performance metrics of the ESM-2 model but also enhanced the ProtBERT model. This method of knowledge injection proves effective in enriching the model embeddings, resulting in improved predictive accuracy and reliability. The cross-attention technique integrates sequence-based features in a manner that enhances the model's ability to discern and predict the properties of anticancer peptides. The ESM-2 with feature integration model not only outperforms existing models in terms of accuracy, F1 score, precision, and specificity, but also exemplifies the efficacy of using cross-attention for knowledge injection.

### D. Ablation Studies of Feature Configurations

Table II summarizes the performance of various feature configurations using the cross-attention method with the ESM-2 embeddings. The key metrics evaluated include accuracy, F1 score, precision, sensitivity, and specificity. This comprehensive analysis helps identify which combination of features contributes most significantly to the model's performance in predicting ACPs.

The results indicate that integrating multiple sequence-based features generally boosts the model's performance. Notably, the combination of all four features achieved the highest accuracy (0.7778) and F1 score (0.7829). This enhancement suggests that each feature captures distinct, crucial aspects of peptide sequences, thereby contributing to a more robust and predictive model.

AAC displayed considerable standalone performance with an accuracy of 0.7661, showcasing its strong foundational relevance in ACP prediction. However, AAC may also lead to false positives due to shared amino acid compositions between ACPs and non-ACPs. This explains why configurations without AAC, such as the trio of DPC, CKS, and k-mer, achieved the highest precision (0.7834) and specificity (0.8012). These features excel in distinguishing non-ACP sequences and minimizing false positives. By removing AAC,

the model focuses on specific features like DPC, CKS, and k-mer, which capture structural details unique to ACPs, thus improving precision and specificity.

It is also worth noting that the incorporation of the CKS feature, in addition to k-mer, using triads (k=3), identifies recurring motifs by classifying amino acids based on their dipole moment and molecular volumes, while CKS captures long-range interactions through the spatial distribution of amino acids grouped by physicochemical properties. As shown in Table II, the combination of CKS and k-mer led to higher precision (0.7640) and specificity (0.7544) compared to either feature alone. This demonstrates the complementary nature of these features in capturing both local sequence patterns and global structural interactions, improving the model's ability to differentiate ACPs from non-ACPs.

DPC was particularly outstanding for achieving the highest sensitivity (0.8011) when used alone, emphasizing its utility in identifying true positive ACP cases. The pairing of AAC and DPC reached the highest sensitivity recorded in the study (0.8129), highlighting the synergistic effect of these features in enhancing model sensitivity.

This ablation study emphasizes the significant impact of integrating diverse configurations of sequence-based features with PLM embeddings through the cross-attention method. The enhanced performance across all metrics when using a combination of features confirms that each provides complementary information, contributing to the overall effectiveness of the model. This integration not only leverages the deep contextual understanding provided by ESM-2 embeddings but also enhances it with specific biochemical insights, leading to more accurate and reliable ACP predictions. Such findings are pivotal for furthering ACP research and could potentially guide the development of more effective anticancer therapies.

### E. Model Interpretation

To investigate the impact of various sequence-based features and PLM embeddings on the model's performance, we employed eXplainable Artificial Intelligence (XAI) to interpret the model and observe the effects of each feature. XAI is a

set of processes and methods that allow human users to comprehend and trust the output of machine learning algorithms. By providing clear explanations, XAI helps in understanding how models make decisions, which is crucial for validating and improving model performance.

We employed the Shapley Additive exPlanations (SHAP) technique [20], which allows us to understand individual predictions from a model and assess the importance of different input features. By applying SHAP [20], we can derive additive feature importance measures, which facilitate the identification of the relative significance of various input features.

We conducted a feature analysis focusing on AAC, DPC, k-mer, CKS, and embeddings, where embeddings are the sequence output from the PLM. We selected the ESM-2 model for SHAP analysis as it returned the best results. For this analysis, we used the SHAP Deep explainer, which treats our model's classification head containing the cross-attention as a black box.

The training dataset was split into two halves. The first half was used to train the SHAP model, while the second half was used to compute the SHAP values. Given that all the features are multi-dimensional, we first normalized them and then reduced their dimensions using t-distributed Stochastic Neighbor Embedding (t-SNE) [21] due to the non-linear nature of the data. We then plotted the dependence and summary plots based on the positive class in Figures 2 and 3.

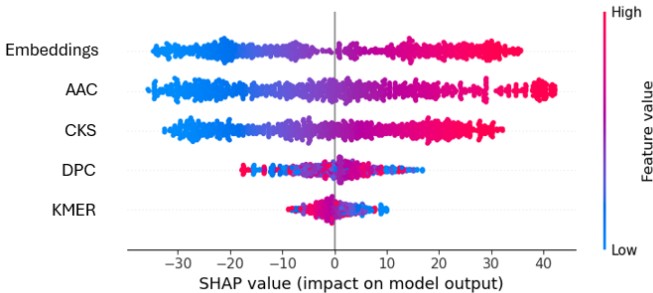

Fig. 2. SHAP values indicating feature importance in predicting ACP.

Figure 2 displays the distribution of SHAP values for each feature, offering a detailed view of how individual feature values influence the prediction of ACP. The X-axis represents the contribution of each feature to the model's prediction. Positive SHAP values indicate that the feature pushes the prediction higher, whereas negative values suggest a downward influence. The color gradient of each feature encodes the magnitude of the feature values, with blue indicating low values and pink representing high values.

From Figure 2, the embeddings exhibit a wide range of SHAP values, from approximately -30 to +40, signifying a substantial influence on the model's output. The distribution indicates that both high and low values of embeddings can have a significant positive or negative impact. AAC also shows a broad range, with SHAP values spanning from -30 to +40. This suggests a notable impact, similar to the embeddings.

CKS displays a narrower range compared to embeddings and AAC, yet remains significant. The SHAP values range between -30 and +30. DPC predominantly influences the model positively, with SHAP values clustered between -20 and +15. K-mer exhibits the least variability in SHAP values, ranging from approximately -10 to +10, indicating a smaller impact relative to the other features.

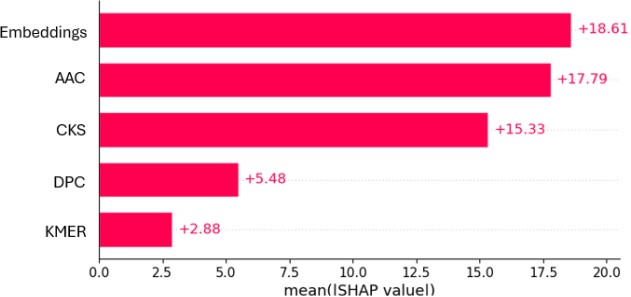

Fig. 3. SHAP ranked feature weights on cross-attention model output.

Figure 3 provides a summary of the average impact of each feature, offering a more concise view of feature importance. The X-axis represents the mean absolute SHAP value, reflecting the average magnitude of each feature's impact on the model predictions. From Figure 3, embeddings emerge as the most influential feature, with a mean SHAP value of 18.61. AAC is the second most impactful feature, with a mean SHAP value of 17.79. CKS holds significant importance with a mean SHAP value of 15.33. DPC displays moderate influence, evidenced by a mean SHAP value of 5.48. K-mer exhibits the least influence among those analyzed, with a mean SHAP value of 2.88. Figures 2 and 3 display the respective features and their impact on the cross-attention model for the positive class. It is interesting to note that in the cross-attention configuration, embeddings and AAC have the highest impact on the model's prediction. This finding is supported by Table II, which shows the high accuracy achieved when only AAC features are used compared to other features.

Other features such as CKS, DPC, and k-mer, while still relevant, exert a comparatively smaller influence. Figure 2 detailed SHAP value distributions highlight the specific contributions of each feature value, while Figure 3 succinctly summarizes their overall importance. By leveraging SHAP to explain our model's predictions, we gain valuable insights into how each feature contributes to the model's decisions. This not only helps in validating the model's behavior but also in identifying potential areas for improvement.

## V. DISCUSSION AND CONCLUSION

Our study focussed on predicting anticancer peptides by integrating sequence-based features with protein language model embeddings. In order to fuse the different features, we used a cross-attention mechanism. By using two PLMs, namely ESM-2 and ProtBERT, we demonstrated that our method enhances the ACP predictions by the PLMs alone.

SHAP analysis further reveals that PLM embeddings and amino acid compositions are the most influential features, with mean SHAP values of 18.61 and 17.79, respectively. While embeddings capture relationships within protein sequences, sequence-based features provide complementary biochemical insights, enhancing the model's overall predictive capability.

The feature integration for ESM-2 models, using cross-attention mechanism achieved the highest accuracy (77.8%) in ACP prediction, outperforming baseline models like CNN, Bi-LSTM, LightGBM, ESM-2, and ProtBERT. It also showed superior F1 score (78.3%), precision (76.5%), and specificity (75.4%), demonstrating its robustness in identifying true positive ACPs while maintaining a low false positive rate. These metrics indicate that the cross-attention mechanism effectively integrates diverse features, enhancing the model's ability to detect intricate patterns indicative of anticancer activity.

Ablation studies validate the importance of integrating multiple sequence-based features. The combination of all four features (AAC, DPC, CKS, and k-mer) resulted in the highest performance in terms of accuracy and F1 score with 77.8% and 78.3% respectively. The combination of AAC and DPC alone achieved the highest sensitivity (81.3%), highlighting their critical role in detecting true positive cases. Conversely, DPC, CKS, and k-mer achieved the highest precision (78.3%) and specificity (80.1%), demonstrating their effectiveness in correctly identifying non-ACP sequences, minimizing false positives.

Future research could explore several avenues to further enhance ACP prediction models. Firstly, incorporating additional biochemical features or larger external biological databases could provide even more comprehensive insights into peptide characteristics. Secondly, extending the cross-attention mechanism to include other types of biological data, such as structural or functional annotations, may further improve the model's performance. Lastly, implementing advanced interpretability techniques beyond SHAP, such as integrated gradients [22] or attention flow analysis [23], could provide deeper insights into the model's decision-making process, facilitating more transparent and trustworthy predictions.

In conclusion, our study shows the potential of combining PLM embeddings with sequence-based features using cross-attention mechanisms to enhance ACP prediction. This approach not only achieves high predictive accuracy but also offers valuable interpretability, paving the way for more effective and reliable computational tools in anticancer peptide research.

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
