# OpenReview forum: "Enhancing Protein Language Models with Feature Integration for Anticancer Peptide Prediction"
_IEEE.org/EMBS/BHI/2024/Conference — IEEE BHI'24_

### Official Review · Reviewer_YfGV · 2024-08-07
**Reviews of Submission99**

**Overall Rating:** 6
**Confidence:** 3

**Other Quality Metrics:**

(a) Clarity of writing --- good
(b) Clinical Significance --- good
(c) Methodological Novelty --- good
(d) Experiments and Results --- good

**Questions For The Authors:**

See the weakness.

**Strengths:**

1. The integration of sequence-based biochemical features with deep learning models (PLMs) through a cross-attention mechanism is an advancement in peptide prediction.
2. The use of SHapley Additive exPlanations (SHAP) to analyze the contribution of different features enhances the interpretability of the model. This analysis is helpful in understanding which features are most influential in the prediction process.

**Summary Of The Paper:**

This paper addresses the challenge of predicting anticancer peptides (ACPs) using advanced computational models. The study leverages protein language models (PLMs) like ESM-2 and ProtBERT, which are pre-trained on vast datasets to capture complex biological patterns in protein sequences. Recognizing the limitations of PLMs in capturing specific biochemical interactions essential for accurately predicting ACPs, the study enhances these models by integrating four sequence-based features: amino acid composition (AAC), dipeptide composition (DPC), composition of K-spaced amino acid group pairs (CKS), and k-mer sparse matrix through a cross-attention mechanism. This integration aims to provide a richer, more informative representation of peptides, thereby improving the accuracy of ACP predictions. The study reports improvements in predictive performance, with enhanced accuracy and some insights into the biochemical properties of peptides.

**Weaknesses:**

1. The study incorporates four specific biochemical features but lacks an explanation for their selection over other potential features such as Physicochemical Properties, Secondary Structure Propensity, Aromaticity, and Isoelectric Point. Could you clarify the rationale behind these choices?
2. Balancing the impact of raw PLM embeddings with additional sequence-based features can be challenging. How have you managed this aspect during model training?
3. The results indicate varying degrees of improvement from feature integration on ProtBERT and ESM-2, underscoring the significance of selecting highly informative features for each specific PLM. Have you considered using a broader set of features and employing cross-attention mechanisms to automate the feature selection process?

---

### Official Review · Reviewer_BTYt · 2024-08-12

**Overall Rating:** 7
**Confidence:** 4

**Other Quality Metrics:**

(a) Clarity of writing: good

(b) Clinical Significance: great

(c) Methodological Novelty: good

(d) Experiments and Results: great

----------

Please note that the reviewer has read the authors' rebuttal and has updated the overall rating as a result.

**Questions For The Authors:**

1. Regarding Table II, do the authors have any insights why excluding the AAC feature may have a significant benefit in enhancing the specificity? (and also improving the precision).

2. Furthermore, it would be helpful if the authors could elaborate on how the incorporation of CKS feature in addition to the k-mer feature enhances the predictive performance (w.r.t the results in Table II).

**Strengths:**

1. Overall, the paper is well-written, where the technical details are clearly presented and the advantages of the proposed method are sufficiently backed up by comprehensive performance analysis results.

2. Especially, results in Table 1 clearly show the advantages of integrating ESM-2 embeddings with sequence-based additional features compared to various baseline methods.

3. The ablation study results are summarized in Table II, which demonstrate the benefits of incorporating all four suggested features (AAC, DPC, CKS, k-mer features) in enhancing the overall prediction accuracy.

4. SHAP analysis provides additional insights regarding the efficacy of the additional sequence-based features.

**Summary Of The Paper:**

This paper proposes a method for improving anticancer peptide prediction by integrating the embeddings from protein language models (PLMs) with additional sequence-based features.

**Weaknesses:**

1. It is not clear what motivated the authors to select the four sequence-based features. Please provide a clear justification/motivation.

2. While the authors provide some explanations regarding the potential benefits of the incorporated sequence-based features, these explanations are very high-level and not necessarily useful.
For example, how does including AAC help characterize "structural" and "functional" attributes?
Similarly, how does DPC offer insights into "structural" and "functional" aspects of anticancer peptides?

3. In principle, it appears that k-mers might be able to capture the info that CKS attempts to capture.
Please elaborate on why incorporating separate CKS feature in addition to the k-mer features may provide additional value.

4. Why was k=3 chosen for the k-mers? Please justify. How sensitive would be the overall performance on the choice of k?

5. Figure 1 is blurry. Please use a high-resolution image.

---

### Decision · Program_Chairs · 2024-09-23

Accept